# Creation of Two-Line Fragrant Glutinous Hybrid Rice by Editing the *Wx* and *OsBADH2* Genes via the CRISPR/Cas9 System

**DOI:** 10.3390/ijms24010849

**Published:** 2023-01-03

**Authors:** Yahong Tian, Yin Zhou, Guanjun Gao, Qinglu Zhang, Yanhua Li, Guangming Lou, Yuqing He

**Affiliations:** National Key Laboratory of Crop Genetic Improvement and National Center of Plant Gene Research, Hubei Hongshan Laboratory, Huazhong Agricultural University, Wuhan 430070, China

**Keywords:** *Wx*, *OsBADH2*, CRISPR/Cas9, two-line hybrid rice, 2-acetyl-1-pyrroline aroma, quality

## Abstract

Global food security has benefited from the development and promotion of the two-line hybrid rice system. Excellent eating quality determines the market competitiveness of hybrid rice varieties based on achieving the fundamental requirements of high yield and good adaptability. Developing sterile and restorer lines with improved quality for two-line hybrid breeding by editing quality genes with clustered regularly interspaced short palindromic repeat (CRISPR)/Cas9 is an efficient and practical alternative to the lengthy and laborious process of conventional breeding to improve rice quality. We edited *Wx* and *OsBADH2* using CRISPR/Cas9 technology to produce both homozygous male sterile mutant lines and homozygous restorer mutant lines with Cas9-free. These mutants have a much lower amylose content while having a significantly higher 2-acetyl-1-pyrroline aroma content. Based on this, a fragrant glutinous hybrid rice was developed without too much effect on most agronomic traits. This study demonstrates the use of CRISPR/Cas9 in creating two-line fragrant glutinous hybrid rice by editing the components of the male sterile and the restorative lines.

## 1. Introduction

The most significant physical and chemical factor that impacts rice quality is the amylose content in rice grains [1,2,3]. *Wx* is important for the regulation of rice quality (including appearance quality and eating and cooking quality (ECQ)) and encodes granule-bound starch synthase I [4,5,6]. The abundant natural allelic variations confer the extensive variation of amylose content and rice quality among modern cultivated rice [6]. At least 10 distinct functional alleles of *Wx* have been identified, including *Wx^a^*, *Wx^b^*, *wx*, *Wx^in^*, *Wx^op^*, *Wx^mp^*, *Wx^mq^*, *Wx^hp^*, *Wx^lv^*, and *Wx^la^*/*Wx^mw^* [6,7,8,9,10,11,12,13]. Of these, *Wx^lv^* is an ancestral allele derived from wild rice. *Wx^la^*/*Wx^mw^* is a recently identified *Wx* allele derived from intragenic recombination, giving rice good eating qualities and grain transparency. The clustered regularly interspaced short palindromic repeat (CRISPR)/Cas9 system appears to be a popular trend for editing various components of the *Wx* gene to enhance rice quality. Zhang et al. [14] and Xu et al. [15] edited the exon of the *Wx* gene, while Huang et al. [16] and Zeng et al. [17] edited the promoter and 5′UTR intron of the *Wx* gene, respectively. Thus, several mutants with different quality traits were produced. According to Liu et al. [18], the amylose content was significantly increased in rice seeds by editing the first intron of the *Wx* gene. These findings suggest that CRISPR/Cas9-mediated gene editing of suitable target sites can produce ideal amylose content and quality breeding materials.

The eating quality of rice is significantly influenced by several factors, including aroma, which is mainly controlled by the recessive gene *fgr*/*OsBADH2* [19]. When the *fgr*/*OsBADH2* gene underwent loss of function, it caused the BADH2 protein to lose its ability to catalyze the oxidation of 4-amino-butanal. This led to an accumulation of 4-amino-butanal, which promoted the synthesis of 2-acetyl-1-pyrroline (2-AP), and resulted in the production of fragrance in rice [20]. Shan et al. edited the *fgr*/*OsBADH2* gene in Nipponbare using transcription activator-like effector nuclease (TALEN) technology, and the 2-AP content in the homozygous T_1_ line was significantly increased [21]. Ashokkumar et al. used the CRISPR/Cas9 method to produce novel alleles of *fgr*/*OsBADH2* to induce aroma into the top non-aromatic rice variety ASD16. The phenotype was stably inherited in the T_1_ generation [22]. Any non-aromatic rice variety can be made aromatic by gene editing *fgr*/*OsBADH2*.

Heterosis refers to the superior performance of hybrids over their parents. A typical example of using heterosis is the creation of two-line hybrid rice systems. One of the fundamental components of two-line hybrid rice breeding is the light- and temperature-sensitive male sterile line. Although the adoption and use of two-line hybrid rice have significantly improved rice yield, the quality of hybrid rice is typically subpar. Three efficient gene editing techniques can be utilized to accurately and rapidly modify crop target traits: TALEN technology [23], zinc finger nuclease technology [24], and CRISPR technology [25]. In this study, we edited *Wx* and *OsBADH2* using the CRISPR/Cas9 system to create homozygous mutants of the Zhinong 1S (ZN1S) sterile and the Zhinong 1307 (ZN1307) restorer lines, resulting in fragrant glutinous hybrid rice with better yield performance than its parents and providing new ideas and insights for quality improvement of hybrid rice.

## 2. Results

### 2.1. Creation of Fragrant Glutinous Mutants with Cas9-Free

We developed *wx*-*fgr* double mutants using CRISPR/Cas9 technology on the genetic background of the sterile line ZN1S and restorer line ZN1307 to produce fragrant and waxy hybrid rice (Figure 1A). PAGE combined with Sanger sequencing analysis revealed that ZN1S and ZN1307, respectively, had heterozygous mutations of 9 and 6 distinct mutation types in T_0_ transgenic plants (Appendix A). We PCR-selected Cas9-free plants from transgenic (T_1_–T_2_) segregating families and identified 5 and 6 homozygous mutant T_3_-lines for ZN1S and ZN1307, respectively (Figure 1B). Unlike the translucent endosperm of wild type grains, the endosperm of all the mutants showed milky white similar to that of glutinous rice (Figure 1C). The amylose content of grains from these 11 mutant T_3_-lines was measured, and the results revealed that once the *Wx* gene was mutated, the amylose content in every mutant drastically dropped (Figure 1D). The lowest was the *wx*-*fgr*-S2 mutant line, whose amylose content was 2.2%, which was very close to that of wild glutinous rice. Among them, the amylose content in grains of ZN1S mutants ranged between 2–4%. Amylose content ranged between 2.5–3.5% in ZN1307 mutants, with *wx*-*fgr*-R2 having the lowest value at 2.6% (Figure 1D). Additionally, we used the potassium hydroxide method to conduct a sensory evaluation of these mutants’ aromas. The results revealed that the mutant brown rice from both materials could generate a light aroma of rice (Figure 1E). Next, we assessed the 11 homozygous mutant T_4_-lines for ZN1S and ZN1307′s quality traits. All mutants’ amylose content (for convenience, the abbreviation AC is used below for amylose content) was significantly lower than the corresponding wild type, consistent with the results of T_3_ generation, while their 2-AP and GC were significantly higher (Figure 1F–I). Rapid viscosity analysis (RVA) was used to evaluate the starch quality [26]. Compared to WT-S, the viscosity indexes of the remaining three mutants from ZN1S were lower than those of *wx*-*fgr*-S1 and *wx*-*fgr*-S4 (Figure 1J). Different from mutants from ZN1S, all mutants from ZN1307 showed similar viscosity indexes, but these were obviously less than for WT-R (Figure 1J).

Furthermore, we looked at the agronomic traits of these homozygous mutant T_4_-lines for ZN1S and ZN1307 (Appendix A). Only specific agronomic traits of some mutants were altered, such as plant height in *wx*-*fgr*-S5, number of effective tillers in *wx*-*fgr*-S1 and *wx*-*fgr*-S3, number of primary branches in *wx*-*fgr*-S2, and grain number per panicle in *wx*-*fgr*-S1 and *wx*-*fgr*-S5, etc. Similar to those of ZN1S mutants, most agronomic traits did not change significantly in ZN1307 mutants; only certain mutants showed changes in specific agronomic traits, such as the number of primary branches in *wx*-*fgr*-R1 and *wx*-*fgr*-R2, the setting rate in *wx*-*fgr*-R3, the 1000-grain weight in *wx*-*fgr*-R1 and *wx*-*fgr*-R5. The only noteworthy thing is that all ZN1307 mutants had varying degrees of decreased yield per plant.

### 2.2. Creation of Fragrant Glutinous Hybrid Rice

*wx*-*fgr*-S2 with the lowest amylose content among the ZN1S sterile mutants and *wx*-*fgr*-S4 with similar agronomic traits to the wild-type were selected as recipients. In contrast, *wx*-*fgr*-R2 with the lowest amylose content among the ZS1307 restorer mutants and *wx*-*fgr*-R6 with the least impact on yield per plant were selected as donor parents to obtain four types of hybrid rice, namely *wx*-*fgr*-Z22 (*wx*-*fgr*-S2/*wx*-*fgr*-R2), *wx*-*fgr*-Z42 (*wx*-*fgr*-S4/*wx*-*fgr*-R2), *wx*-*fgr*-Z26 (*wx*-*fgr*-S2/*wx*-*fgr*-R6), and *wx*-*fgr*-Z46 (*wx*-*fgr*-S4/*wx*-*fgr*-R6). We identified the four transgenic hybrid rice’s primary quality features. Similar to ZN1S sterile mutants and ZS1307 restorer mutants, the grain endosperms produced by all hybrid combinations had low transparency and milky white appearance, obviously different from that of WT-S and WT-R (Figure 1C). Iodine staining results showed that refined rice grains from hybrid rice had lighter coloration in transverse sections than WT-S and WT-R (Figure 2A). Correspondingly, the four transgenic hybrid rice variants have much less amylose than their two parental strains (Figure 2B). In particular, *wx*-*fgr*-Z42 had an amylose content of 1.76%, which was on par with wild-type waxy rice. The four transgenic hybrid rice strains had an aroma substance 2-AP level that was significantly higher than that of the sterile line ZN1S while not comparable to that of the restorer line ZN1307 (Figure 2C,D). However, rice flour from the four transgenic hybrid rice variants had a worse RVA curve pattern than that from thetwo parents, WT-S and WT-R, while having a greater gel consistency than the wild type (Figure 2E,F). In addition, we also investigated the main agronomic traits of the transgenic hybrid rice (Appendix A). In general, hybrid rice showed obvious superparent advantage in plant height and panicle length, but no obvious changes in other traits, except for specific traits of several mutants, such as the grain weight per panicle in *wx*-*fgr*-Z46, the setting rate in *wx*-*fgr*-Z26 and *wx*-*fgr*-Z46 and the yield per plant in *wx*-*fgr*-Z26.

## 3. Discussion

Although hybrid rice’s cooking and eating quality have somewhat improved recently, they still fall short of high-grade and high-quality conventional rice. Incorporating the *wx* mutation into varieties with low initial AAC levels led to further reductions in AAC; however, these effects had little to no impact on the desired gelatinization traits, amylopectin structure types, or the major agronomic traits [27]. Therefore, introducing the *wx* mutation into rice varieties with low baseline AAC levels is a feasible strategy for increasing the ECQ of rice. In this study, the wild-type sterile line ZN1S and the restorer line ZN1307 had initial amylose contents of about 10% (Figure 1D,F), which were not very high. All of the mutants’ amylose contents were reduced to 2–4% levels by Cas9-mediated gene editing, along with some modifications to their gelatinizing properties (Figure 1). Notably, sterile line mutants and restorer line mutants have gel consistency that is significantly higher than that of the corresponding wild type (Figure 1I). However, different from the great changes in ZN1S and ZN1307 mutants, only *wx*-*fgr*-Z26 had significantly higher gel consistency in hybrid rice, but only between the two parents (Figure 2F). The viscosity curve of the sterile line mutants varied in degree from the wild type, whereas all restorer line mutants and hybrid rice had softer pasting properties (Figure 1J). These results indicate that the heredity of quality traits in hybrid rice may not be determined by simple additive effect. In addition, it may not be enough to manipulate only *Wx*, the main quality gene. Combining with other quality genes, even some quality genes with minor effects, may be needed to finally improve rice quality.

Overall, our study shows how to directly alter high-quality genes of interest in elite sterile and restorer lines to produce enhanced hybrid rice with the potential for commercialization.

## 4. Materials and Methods

### 4.1. Plant Materials and Growing Conditions

ZN1S is a photoperiod-sensitive genic male sterile rice variety, whereas ZN1307 is an indica restorer rice variety with high quality and high yield, and both carry a *Wx^la^* allele. Plants were grown in the field throughout the regular rice growing season at Huazhong Agricultural University’s experimental station, Wuhan (Hubei), and Lingshui (Hainan). The seeds were sown on May 15 every year, and the seedlings (25 or 30 days old) were transplanted onto the field with a single plant spacing of 16.5 cm and 26.4 cm between rows. The field was managed using customary agricultural practices. At the end of the third stage of panicle differentiation, ZN1S male sterile lines—whose fertility transformation temperature was 24 °C—were generally transferred to a cold water pool (22.5 °C), then after 15 days of treatment from the cold water pool to the field.

### 4.2. Design of the Wx and OsBADH2 Target Sites and Construction of the CRISPR/Cas9 Double-Targeting Vector

We designed target sites at bases 14–33 of the third exon of the *Wx* gene and bases 122–141 of the fourth exon of the *OsBADH2* gene, respectively, to produce waxy and fragrant transgenic lines. The online tool CRISPR GE was used to design the target sites [28]. The target sequences for the *Wx* gene were GGGTCATGGTGATCTCTCCTCGG and for the *OsBADH2* gene ATCAACCCAACTACACCGATAGG. The U6-sgRNA expression cassette encoding the 20 nucleotides (nt) *Wx* target sequence was amplified by polymerase chain reaction (PCR) and ligated into the pCXUN vector, which *Kpn I* digested to create the pCXUN-U6-sgRNA intermediate vector. The 20 nt *OsBADH2* target sequence from the U3-sgRNA expression cassette was then amplified by PCR and ligated into the pCXUN-U6-sgRNA intermediate vector, digested by *Sac I* to create the final vector pCXUN-U6-U3-sgRNA. The constructed CRISPR/Cas9 final vector was introduced into the ZN1S and the ZN1307 receptor using *Agrobacterium tumfaciens*-mediated genetic transformation [29]. The primer sequences used to construct the vector are listed in Appendix A.

### 4.3. Molecular Characterization of the Mutant Plants

Genomic DNA was extracted from seedling leaves using the sodium dodecyl sulfate method [30]. PCR amplification was performed using primer pairs that produce amplicons containing the target sites. The amplified products were then sequenced using the Sanger sequencing method and assembled using the software SeqMan from the Lasergene package. We compared the amplicon sequences generated from the corresponding wild-type and real transgenic templates to identify mutations. Polyacrylamide gel electrophoresis (PAGE) was used to determine the homozygosity/heterozygosity for a mutant individual. Indel primer pairs that produce an amplicon, including the target sites, were created. Agarose gel electrophoresis was used to identify Cas9 in each individual. The relevant PCR primers for these steps are listed in Appendix A.

### 4.4. Evaluation of Rice Quality

Amylose content and gel consistency were measured using the previously described method [31]. Ten full grains of brown rice were placed in a Petri dish for the sensory evaluation of rice fragrance. Following this, 10 mL of 1.7% potassium hydroxide solution was added, and the dish was covered and left at room temperature for 10 min. The Petri dishes were then opened one by one, and at least three subjects with a normal sense of smell were requested to assess and average each sample. Gas chromatography-mass spectrometry (GC-MS) was used to determine the content of flavor ingredient 2-AP. The extract was added after the brown rice was ground into a powder. The final sample was injected into the Agilent6890 GC connected to the HP5973 MS detector (Agilent Technologies, Palo Alto, CA, USA) for detection. MS database and retention time were used to identify the volatile compounds. Internal standard methylene chloride. 2, 4, 6-trimethyl pyridine was injected at a concentration of 0.4585 ng/µl, and the injection volume was 1 µl. A previous study has outlined the detailed steps [32].

### 4.5. Investigation of Agronomic Characters

In the paddy field, plant height, grains per panicle, panicle length, effective panicles, primary branches, and secondary branches were measured. Grain number per panicle, grain weight per panicle, setting rate, 1000-grain weight and yield per plant were investigated indoors. Seeds were collected for each plant and dried at 37 °C for two weeks to determine the yield per plant. Using the SC-A grain analysis system (Wseen company, Hangzhou, China), 1000-grain weight was measured. Three similar-sized panicles from each plant were selected, dried at 37 °C for two weeks, and the average value was then calculated to determine the number of grains and grain weight per panicle.

### 4.6. Performance of the F_1_ Hybrids Obtained from ZN1S × ZN1307

ZN1S mutant lines (*wx*-*fgr*-S2 and *wx*-*fgr*-S4) and ZN1307 restorer lines (*wx*-*fgr*-R2 and *wx*-*fgr*-R6) were crossed in pairs. To avoid unforeseen pollination, the panicles were covered with brown paper bags. The F_1_ seeds were harvested 30 days after the grains matured, and the F_1_ hybrids were sown in the field plots. The experiment used a replicative complete block design with 36 plants in each plot that was replicated twice.

### 4.7. Microscopy

In the iodine staining experiment, the mature seeds were first processed into milled rice, and the milled rice was cut with a sharp blade after being stained with iodine solution, and then observed and photographed with a stereo fluorescence microscope (SMZ25, Nikon, Tokyo, Japan). The appearance of milled rice was directly observed and photographed by a stereo fluorescence microscope (SMZ25, Nikon, Tokyo, Japan).

### 4.8. Statistical Analysis

The various diagrams in this study were drawn using GraphPad Prism 8. One-way analysis of variance and Duncan multiple comparisons were used to analyze differences between different groups using IBM Statistical Package for Social Sciences v16.0 software. Microsoft Excel 2016 was used to perform the preliminary processing and analysis of phenotypic data.

## Figures and Tables

**Figure 1 ijms-24-00849-f001:**
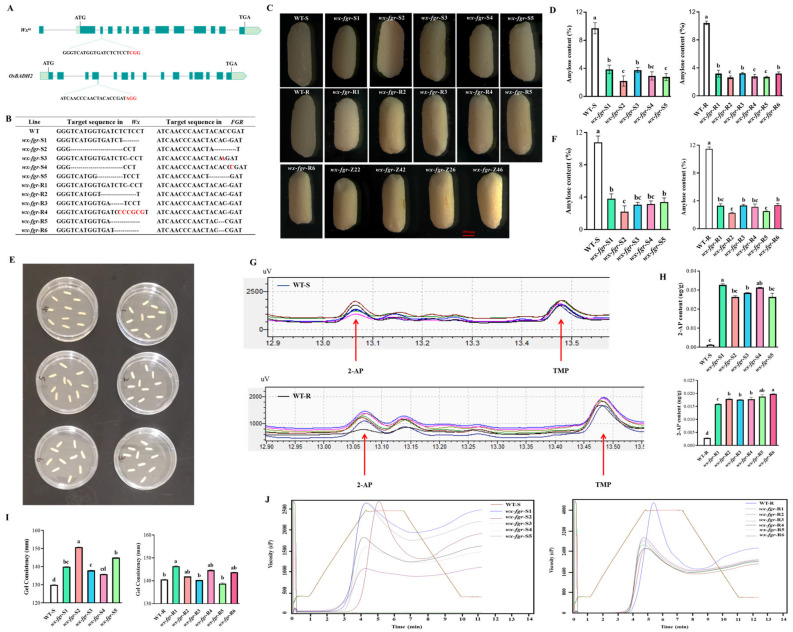
Improvement of rice grain quality by editing of the *Wx^la^* and *OsBADH2* using CRISPR/Cas9. (**A**) Schematic diagram of the targeted sites in *Wx* and *OsBADH2*. The protospacer-adjacent motifs (PAMs) are shown in red. (**B**) Mutations in the edited T_3_ lines. Inserted bases are marked in red, and missing bases are indicated by short dashed lines. (**C**) Morphology of milled rice from ZN1S, ZN1307 and the F_1_ hybrid ZN1S mutants × ZN1307 mutants. WT-S represents the wild type of sterile line ZN1S and WT-R represents the wild type of restorer line ZN1307. Scale bars, 1000 µm. (**D**) The amylose contents determined by iodine colorimetry of ZN1S, ZN1307 and their T_3_ generation homozygous mutant lines. (**E**) Sensory evaluation of the aroma of ZN1S, ZN1307 and their mutants by potassium hydroxide method. (**F**) The grain amylose contents determined by iodine colorimetry of ZN1S, ZN1307 and their T_4_ generation homozygous mutant lines. (**G**) Total ion chromatograms (TIC) of 2-AP and TMP in the grains of ZN1S, ZN1307 and their T_4_ generation homozygous mutant lines. (**H**) 2-AP content in grains of ZN1S, ZN1307 and their T_4_ generation homozygous mutant lines. 2, 4, 6-trimethyl pyridine (TMP) was used as the internal standard. (**I**) Gel consistency. (**J**) Rapid visco analysis profiles of grain starches of ZN1S, ZN1307 and their T_4_ generation homozygous mutant lines. cP (centi Poise), viscosity unit. Error bars are means ± SD (*n* = 3). Samples without the same letter show significant difference by Duncan’s test (*p* < 0.05).

**Figure 2 ijms-24-00849-f002:**
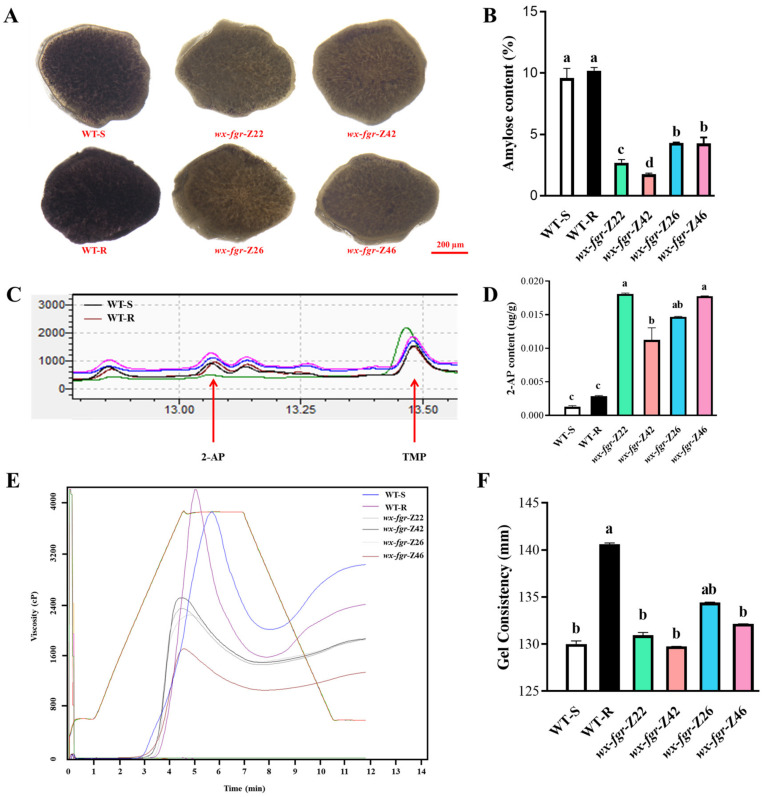
Creation of two-line fragrant glutinous hybrid rice. (**A**) Microscopic observation of iodine-stained endosperm. WT-S represents the wild type of sterile line ZN1S and WT-R represents the wild type of restorer line ZN1307. Scale bars, 200 µm. (**B**) The amylose contents determined by iodine colorimetry of ZN1S, ZN1307 and the F_1_ hybrid ZN1S mutants × ZN1307 mutants. (**C**) Total ion chromatograms (TIC) of 2-AP and TMP in the grains of ZN1S, ZN1307 and the F_1_ hybrid ZN1S mutants × ZN1307 mutants. 2, 4, 6-trimethyl pyridine (TMP) was used as the internal standard. (**D**) 2-AP content. (**E**) Rapid visco analysis. cP (centi Poise), viscosity unit. (**F**) Gel consistency. Error bars are means ± SD (*n* = 3). Samples without the same letter show significant difference by Duncan’s test (*p* < 0.05).

## Data Availability

For materials, please contact the corresponding author’s email address.

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
