# Peer review of "Creation of Two-Line Fragrant Glutinous Hybrid Rice by Editing the Wx and OsBADH2 Genes via the CRISPR/Cas9 System"

_ijms, 2023, doi:10.3390/ijms24010849_

Round 1
Reviewer 1 Report
I have following concerns for the authors to improve the manuscript during their revision.
1. The double mutation lines showed generally lower AC contents and higher 2-AP contents, but there are still some different among the lines. Furthermore, several agronomic traits such as plant height did show some different. Is it possible these variations are related to potential off-target mutations?
2. The authors use RZN1307 in lines 134, 178 and 238, but instead ZN1307 is stated in the other places. In Figures, WT-S and WT-R are used without annotation. Hope these misleading statements can be settled.
3. Figure 2A is not clear enough. The text in red should not place on the endosperm, instead it is better put underneath.
Reviewer 2 Report
The manuscript “Creation of two-line fragrant glutinous hybrid rice by editing the Wx and OsBADH2 genes via the CRISPR/Cas9 system” described the utilization of CRISPR/Cas9 system targeting Wx and OsBADH2 genes in the sterile line ZN1S and restorer line ZN1307 to produce hybrid rice variety. The aromas and the amylose content of the rice grain were examined. Field performance including plant height, panicle length, number of tillers, number of primary branches, grain number per panicle, grain weight per panicle, setting rate, 1000-grain weight and yield per plant of the Cas9-free edited rice was also evaluated in this study.
Minor suggestions/Corrections:
-
Page 1, Line 13 and Line 34, CRISPR should be acronym of (clustered regularly interspaced short palindromic repeats)
-
Page 2, Line 48, TALEN should be acronym of “transcription activator-like effector nuclease”
-
Page 2, Line 85, what does “AC” mean, amylose content, please specify before using that acronym?
-
Page 3 and 5, Figure 1I and 2F. Gel consistency(mm), please correct the word gel instead of cel consistency in the y-axis label
-
Page 5, Figure 2A, it would be better to change the red font color to other color since the label is not clear to the readers if written in red color through the rice endosperm.
-
Page 6, Line 196 and 199, SacI, please italicize the enzyme name, similarly, KpnI, should also be italicized
-
Page 6, Line 210, suspected transgenic templates, what does this mean? authors should use the specific primers to determine the real transgenic plants.
Other comments/Major concerns:
-
Are there any potential off target effects of the guide RNA designed for the Wx and OsBADH2 genes? please provide relevant evidence.
-
For the PCR approach to determine the Cas9-free plants, how many pairs of primers were used for screening the transgene? Please include in the supplemental table. Please indicate the primers usages or description should be included in the supplemental table 1. Did the authors perform the whole genome sequencing of the Cas9-free rice plants for detecting the transgene insertion before the field trial.
-
Page 3, Line 116, in the results section, the authors mentioned that ZN1307 rice mutants had varying degrees of decreased yield per plant, what would be the potential explanation?
-
For examination of the Gel Consistency(GC) of the rice grains, it seems there is not much difference between WT-S and the mutant lines as shown in Figure 2F when compared to their parents as shown in Figure 1I ? Did the authors have any reasonable explanations for that?
Round 2
Reviewer 2 Report
I did not see any changes were made in the main text for the two comments I pointed out in the first review
1. Page 1, Line 15-16 and Line 36-37, CRISPR should be acronym of (clustered regularly interspaced short palindromic repeats)
2. Page 2, Line 51-52, TALEN should be acronym of “transcription activator-like effector nucleases”
